# Overall time spent by clients from entry to exit and associated factors in out-patient departments in public hospitals of Jimma Zone southwest, Ethiopia

**Zebader Walle[1]\*, Frehiwot Worku[2], Yibeltal Sraneh[3], Dejenie Melese[3], Tilahun Fufa[3], Elias Ali Yesuf[3,4], Gete Berihun[5]**

1 Department of Public Health, College of Health Sciences, Debre Tabor University, Debre Tabor, Ethiopia, 2 Department of Public Health, St. Paul Millennium Medical College, Addis Ababa, Ethiopia, 3 Department of Health Policy and Management, Faculty of Public Health, Jimma University, Jimma, Ethiopia, 4 CIH[LMU] Center for International Health, Ludwig-Maximilians-Universität, München, Germany, 5 Department of Environmental Health, College of Medicine and Health Sciences, Wollo University, Dessie, Ethiopia

\* zedwalle123@gmail.com

## Abstract

### Background

The overall time refers to the amount of time a patient spends in a health care facility, from the time he or she enters to the time he or she leaves. As a result of the imbalance between supply and demand, waiting times occur. Ethiopian hospitals are being reformed to improve the quality of care they provide. The time a patient spends in the hospital is one of the most important indicators of quality of care, as it provides insight into customer satisfaction and provider success. However, the overall time patients spend in hospitals was not studied.

### Objective

The study aimed to assess the overall time spent by clients from entry to exit and associated factors in the outpatient departments of Jimma zone hospitals.

### Methods

An institution-based cross-sectional study was conducted. Patients from outpatient units at Jimma zone public hospitals participated in the study from March 15 to May 17, 2018. Data were collected using a time and motion tool coupled with an interviewer-administered structured questionnaire on 249 samples. Participants in the study were selected using the consecutive sampling method. Overall time, in terms of waiting and service times at each section unit, and the relationship of socio-demographic and clinical factors with overall time was the main outcome variables. Data were analyzed using descriptive and linear regression analysis. Simple linear regression analysis was used to determine the relationship between the dependent and explanatory variables. Variables were considered significantly associated with the overall time if they had a p-value of less than 0.05 at the 95% confidence interval (CI).

**Data Availability Statement:** All relevant data are within the manuscript and in the Supporting information.

**Funding:** The authors received no specific funding for this work.

**Competing interests:** The authors have declared that no competing interests exist.

**Abbreviations:** FNA/C, Fine needle aspiration/ Cytology; GOPD, General out-patient Department; IOM, Institute of Medicine; JUMC, Jimma University Medical Center; OPD, Out-patient Department.

## Result

The overall response rate was 94.8%. Overall, patients spent a median time of 342.5 minutes. Patients spent 12.7% of the total time as service time and 86% of the time waiting for care. The longest overall times were spent in the laboratory (170 minutes), imaging (95 minutes), other diagnostic units (84 minutes) and examination (83 minutes). The average overall time was increased by 52.03 minutes (95%CI 21.65, 82.412), 4.65 minutes (95%CI 3.983, 5.324), and 96.43 minutes (95%CI 52.076, 140.787) when the patient was referred, the number of patients at the queue was increased by one unit, and patients who had other diagnostic tests performed respectively with $P < 0.005$ & adjusted $R^2 = 0.522$.

## Conclusion and recommendations

The majority of patients stayed for a longer period. Most time was spent waiting for services, particularly in the examination, laboratory, and imaging units. This is strongly related to high patient load, an absence of some services, being referred patients, and patients who had other diagnostic tests. To reduce the number of patients in the queues, hospitals should work hand in hand with the Ministry of Health to enforce policies that are understood and adopted by all workers in the lower healthcare facilities. And hospital administrators are working to strengthen the triaging system to screen patients with minor illnesses. This is because most patients with minor illnesses queue with those with more complicated illnesses. Finally, we recommend that researchers conduct further research on service quality.

## Introduction

Overall time refers to the time a patient spends in a health care facility, from the time he or she arrives to the time he or she leaves the facility, whereas waiting time refers to the time a patient spends at each service delivery site as he or she waits for the required care from the health service provider [1]. During medical treatment, service time refers to the time spent by patients on registration, routine doctor consultations, laboratory/diagnostic tests, procedures, and drug dispensing [2].

As a result of the imbalance between supply and demand, waiting times occur. A queue occurs when demand exceeds supply. Moreover, if supply does not respond to changes in demand, it may be difficult to improve wait times over time [3–5].

To improve clinic efficiency, healthcare organizations can use wait-time and service-time studies to evaluate the effectiveness of individual clinic sessions, design new clinics, improve clinic patterns, and identify personnel needs [6]. When patients check into hospitals, they often face long waits in the waiting areas. The quality of the waiting experience and service time are highly related to patient satisfaction with the care received [7, 8]. There are many dimensions to the efficiency and effectiveness of outpatient services, but one of the most common complaints from patients is excessive waiting time. Excessive wait times are a losing strategy in which patients lose important time, hospitals lose patients and reputations, and staffs are tense and stressed. The most important aspects that determine patient and customer satisfaction are wait time and service time. From the patient's perspective, longer waiting time increases indirect costs. It can increase patients' irritation and reduce their sense of control [9–11].

Long waiting times are common in many developed and developing countries. Because of the complexity of the causes, limited resources, and unpredictable increases in demand, this problem may be difficult to tackle [12].

Socio-demographic characteristics (gender, age, educational status, residence, and occupation), high patient load, patient arrival time, inadequate appointment schedule, type of diagnosis, and type of investigation were some of the factors that contributed to overall time [8, 1, 11, 13, 14].

Long wait times are common in outpatient facilities, and this difficulty contributes to a variety of public health issues, such as hampered access to care; disruptions in hospital work patterns, including health service delivery; efficiency, quality, transparency, and accountability; and patient dissatisfaction [15–17]. As a result, a Citizen Charter, a novel approach to public management, was created and is being implemented to encourage service providers to be responsive and to teach residents about their service entitlements, standards, and rights [18]. The Jimma zone hospitals citizen charter states that the time for registration, general outpatient department (GOPD), antenatal care (ANC) and postnatal care (PNC), pediatric OPD, ophthalmic OPD service and dental OPD service are 5 minutes, 45 minutes, 20 minutes, 20 minutes, 25 minutes and 20 minutes respectively [19].

The length of time a patient spends in the hospital is one indicator of the quality of services. Nevertheless, in October 2016, Jimma Zone hospitals developed a citizen charter to ensure timely care in Jimma Zone hospitals. The total amount of time patients spend in Jimma Zone hospitals, from the time they enter to the time they leave, is not assessed. Therefore, this study was used to assess the total time patients spend in Jimma Zone OPDs from entry to exit, as well as the associated factors.

## Methods

### Study design, settings and periods

A facility-based cross-sectional study was conducted from March 15 to May 17, 2018, in public Hospitals in Jimma Zone southwestern Ethiopia. There are eight governmental hospitals, two private hospitals, and 120 health centres in the Zone. Out of the eight government hospitals, one is a referral hospital, Jimma University Medical Center (JUMC), three are general hospitals, and four are primary hospitals. JUMC serves a total population of 15 million people. The centre has 160, 000 outpatients and 20, 000 inpatients to serve annually. It provides services to a diverse population from three regional states, namely Oromia, Southern Nations, Nationalities and Peoples, and Gambella. It provides four main services namely:—clinical services, laboratory and diagnosis services, facility services and private wing services. Agaro general hospital provides dental, ophthalmic, medical, surgical, gynecological, obstetric and pediatric services. (Source HR, Plan, and Program Office).

### Study populations and exclusion criteria

Patients who received care at outpatient departments of selected public hospitals in the Jimma zone were included, while those who were critically ill (labeled as an emergency case), mentally ill patients who were violent, and patients who came for repeat medications, investigations, or procedures without seeing a doctor were excluded.

### Sample size determination and sampling procedures

Since the outcome variable was continuous and wanted a measure of the meantime, the sample size was calculated by using a T-test. By using "WINPEPI" software at a 5% significance level

and 80% power the required sample size was 249 by adding a 10% non-response rate. A simple random sampling technique was used to select the hospitals. From the eight public hospitals, three hospitals (JUMC, Agaro general hospital, and Seka primary hospital) were selected. Based on the number of outpatient flows in each selected hospital, proportional allocation (179 from JUMC, 45 from Agaro general hospital, and 25 from Seka primary Hospital) was applied to select the participants.

## Operational definition

**Waiting time:** was the time measured in minutes that a patient had to wait at registration, triage, consultation, laboratory, other diagnostic units, and pharmacy to receive a service in the OPD of public hospitals in Jimma zone.

**Servicsime time:** was the time a patient spent with a health worker for registration, consultation, laboratory tests, other procedures, and dispensing of medicines in the OPD of public hospitals in the Jimma zone, measured in minutes.

**Overall time:** was calculated by subtracting the time of the patient's arrival from the time of leaving the OPD. If the cumulative patient waiting time was greater than or equal to 120 minutes, this was considered a long overall time. However, if the cumulative patient waiting time was less than 120 minutes, it was considered a short overall time.

## Data collection and quality assurance procedures

During data collection, patients in the registration waiting area at the selected hospitals were requested to participate in the study until the required sample size of 249 was reached.

There were two tools for collecting the data, which were adapted by reviewing different literature [1, 8, 9, 11, 13, 14, 16, 20–22]. The first tool was the time and motion tool, in which an independent observation for each unit of service delivery is used to calculate time. That is, a similarly set telephone was used to measure the time spent in each unit in terms of waiting and service time. In addition, when the patient went around the hospital, the tool recorded the number of patients in each unit.

The second tool was a structured and translated (Amharic and Afan Oromo) questionnaire administered by the interviewer. This instrument recorded patient demographic data, previous experience with other health services, as well as the purpose and frequency of visits.

Pre-testing of the questionnaire was done on 5% (13) patients in the outpatient department of JUMC before the study period, completeness and consistency of responses were checked and a correction was made. Data collectors were trained in study concepts, methodology, and the instrument used for one day before the start of the study to ensure that they were familiar with the facilities and knew where each was stationed before the study began. The training was provided by the investigator. The data collectors' time was arranged in the same.

Through independent observation and interviewing, data were collected during the day between 8 a.m. and 6 p.m. on five working days for nine weeks.

The data collection was done by establishing a networking system that allowed a data collector to follow more than two patients at a time in one location. Data collectors observed the entrance at each random time when the patient arrived and recorded the patient's arrival time. Verbal consent was then obtained, the number of patients present before the selected patient was counted, and participants' socio-demographic data were recorded. Then, using mobile phones to track the participants through the service sites and documented the real waiting and service time on the tool. In addition, patients' medical records were reviewed to

see the type of diagnosis. Finally, the exit questionnaire was administered at the last service site.

Seven BSc nurses and two BSc public health supervisors were recruited for data collection and trained for one day in basic data collection techniques and interviewing methods. Data collectors and supervisors spoke the local language. Supervisors monitored the data collection and checked the quality and completeness of the questionnaires.

All questionnaires and time-tracking tools were checked by the investigator for completeness and any errors throughout the days of data collection in the middle of the day and in the evening.

Variables recorded included: the principal diagnosis, which summarized the main diagnosis of diseases in public hospitals, reasons why patients were delayed and reasons why they did not visit other health facilities were summarized to the main reasons that patients' state and recoded for easy analysis. After data entry, cleaning and editing were made to check the accuracy of the entry, explore the entered data for errors, and manage errors.

## Data analysis

The main outcome measures of the study were the following time intervals. (i) The time patients spent at each section of the out-patient unit before getting health care (waiting time); and (ii) the time patients spent with health professionals at each section of the out-patient unit (service time). A composite interval of interest was the time from arrival to leaving the assessment centre (overall time).

All linear regression assumptions were verified. The normality of the distribution was examined using a histogram and a P-P plot. The scatter plot was used to assess linearity. Multicollinearity was determined by assessing the variance inflation factors and tolerance, with values less than 2 and greater than 0.85 indicating no resemblance or independence, respectively. Finally, all residuals and the scatter plot were examined for constant variance/homoscedasticity. As a result, all plots and included points should have the same width.

To summarize the socio-demographic data, descriptive statistical analysis was performed, and a summary of the time spent in contact with a health worker (service time), time spent waiting to see the health worker (waiting time), and overall time were generated and presented in a table and graph using mean, median, and standard deviation.Simple linear regression analysis was carried out, and significant variables with a p-value of less than 0.25 were selected as candidate variables for multivariable linear regressions. However, variables with several cases less than ten were not included in the model. At a significance level of p-value 0.05 with a confidence interval of 95%, multivariable linear regression analysis was done to identify factors that predict overall time. The reduced final model was constructed stepwise using the backward method in descending order in a stepwise way until the model contained only variables with statistical significance differences from their reference variable (P 0.05). The adjusted $R^2$ value was used to test the goodness of the model fit.

## Ethical considerations

The study was performed following the Declaration of Helsinki. Ethical approval was obtained from the Institutional Review Board (IRB) of the Institute of Health Sciences, Jimma University. Permission to conduct this study was obtained from higher officials of the selected hospitals. Written informed consent from all study participants was not possible because of their background and other related factors; we relied on oral (verbal) consent. Approval to use verbal consent was obtained from the Institutional Review Board (IRB), Institute of Health Science, and Jimma University. For the pediatric age group, verbal consent was obtained from

their parents or relatives who were the legal guardians of the children. Before data collection, the aim of the study and the measurement tool was explained to the data collectors and supervisors. The confidentiality of the study was secured by avoiding possible identifiers such as the name of the respondents.

## Result

### Socio-demographic characteristics

Of 249 patients, 236 (94.78%) had a response rate that passed through registration and clinical examination, 132 through triage, 19 went for X-ray, 136 to the laboratory, and 202 through the pharmacy of the assessment centre (Table 1).

### Pre-visit facility characteristics of the patient

Before coming to the study hospitals, 121 (51.3%) patients had visited other health facilities for a similar reason, among these, 65 (53.7%) had visited health centres (Table 2).

### Post-visit facility characteristics of the patient

At the time they entered the study hospitals, 119 (50.4%) had arrived between 8–9 am. During their visit, 125(52.9%) reported they spent a long time in the hospital (Table 3).

### Time spent by patients

**Waiting time.** Overall, patients spend a maximum and minimum time of 557 minutes (9:17hrs) and 5 minutes respectively waiting to be attended by any health worker. This accounts for 86% of the total overall time. The average and median time of total wait was also found to be 213.9min and 222.5 minutes respectively with a standard deviation of ±122.3 minutes. Twenty-five per cent (59) of patients waited more than five hours (Table 4). The mean (SD) waiting time the patient spent in JUMC, Agaro general hospital, and Seka primary hospital was 235.4 ±117.56 minutes, 176.76 ±114.27 minutes and 99.9 ± 97.36 minutes respectively (S1A and S1B Table).

**Service time.** The median and average service time that the patient spent with contact to the health worker was 43.5 minutes and 50.3 minutes with a standard deviation of ± 28.5 minutes respectively. This accounts for 12.7% of the overall time patient spent from entry to exit (Table 5). The mean service time the patient spent in medical, gynecology, ophthalmic and dental OPDs was 25.34 minutes, 29.33 minutes, 17.29 minutes and 37.75 minutes respectively (S2B Table). The mean (SD) service time the patient spent at JUMC, Agaro general hospital and Seka Primary hospital was 50.01±29.6 minutes, 55.37 ±26.79 minutes and 43.9 ± 19.74 minutes respectively (S2A and S2C Table).

**Overall time.** The maximum and minimum overall time spent in OPD of Jimma zone public hospital was 1180 minutes (19:40hrs) and 37 minutes respectively. The average and median time of hospital stay were also found to be 312.3minutes and 342.5 minutes respectively with a standard deviation of ±160.2 minutes. Almost 86% of the overall time was spent as waiting time while around 12.7% accounts for service time (Fig 1).

For pediatric patients, the mean and median overall time was 231.02 minutes and 177.5 minutes respectively with a standard deviation of ±153.9minutes. While for the adult age group, the mean and median overall time was 383.3minutes and 396 minutes respectively with a standard deviation of ±129.15 minutes (Fig 2).

**Table 1. Socio-demographic characteristics of the study participants in OPD of Jimma Zone hospitals, in the southwest, Ethiopia, 2018.**

| Variables | Frequency | Percentage |
|---|---|---|
| **Name of the hospital** | | |
| JUMC | 177 | 75 |
| Agaro general hospital | 38 | 16.1 |
| Seka primary hospital | 21 | 8.9 |
| **Respondent's sex** (n = 236) | | |
| Male | 113 | 47.9 |
| Female | 123 | 52.1 |
| **Age (in a year)** (n = 236) | | |
| ≤14 | 110 | 46.6 |
| 15–29 | 61 | 25.8 |
| 30–44 | 29 | 12.3 |
| 45–59 | 25 | 10.6 |
| >59 | 11 | 4.7 |
| **Residency** (n = 236) | | |
| Jimma town | 120 | 50.8 |
| Out of Jimma town | 116 | 49.2 |
| **Employment status*** (n = 134) | | |
| Student | 28 | 20.9 |
| Unemployed | 47 | 35.1 |
| Self-employed | 37 | 27.6 |
| Formal-employed | 20 | 14.9 |
| Non-formal employed | 2 | 1.5 |
| **Monthly income of patient in ETB****(n = 72) | | |
| ≤500 | 6 | 9.7 |
| 501–1500 | 13 | 21 |
| 1501–3000 | 22 | 35.5 |
| >3000 | 21 | 33.9 |
| **Educational status** (n = 172) | | |
| Unable to read and write | 30 | 17.4 |
| Informal education | 7 | 4.1 |
| Primary school | 71 | 41.3 |
| Secondary school | 30 | 17.4 |
| Tertiary | 34 | 19.8 |
| **Language barrier** (n = 236) | | |
| Yes | 38 | 16.1 |
| No | 198 | 83.9 |

*depend on Ethiopian labor organization that means a person with an age of ≥14years

**ETB = Ethiopian Birr, classification is based on literatures.

## Factors associated with overall time spent

In bivariate linear regression analysis showed that the age of the patient, residency, employment status, income and type of hospital was associated with the overall time the patient spent in Jimma zone public hospitals under the socio-demographic variables (S4A Table).

On the other, under the pre-visit variables, visiting another health facility, the purpose of the visit, and arrival time were factors associated with the overall time (S4B Table). Under

**Table 2. Pre-visit facility characteristics of respondents prior to coming to the outpatient units of Jimma zone hospitals southwest, Ethiopia, 2018 (n = 236).**

| Pre-visit characteristics | Frequency | Percentage |
|---|---|---|
| **Visited other health facility** (n = 236) | | |
| Yes | 121 | 51.3 |
| No | 115 | 48.7 |
| **Type of facility the patient visit**(n = 121) | | |
| Drug shop | 6 | 5 |
| Private clinic | 14 | 11.6 |
| Health center | 65 | 53.7 |
| Gov't hospital | 35 | 28.9 |
| Non-gov't hospital | 1 | |
| **Reason for failure to visit** (n = 115) | | |
| Appointment patient | 4450 | 38.3 |
| For better service | 11 | 43.5 |
| Private not found at any time | 6 | 9.6 |
| The cost of private is high | 4 | 5.2 |
| Health centers do not give full service | | 3.5 |
| **Frequency of visit** (n = 236) | | |
| New attend | 106 | 44.9 |
| Repeat attend | 130 | 55.1 |
| **Purpose/type of visit** (n = 236) | | |
| Review/appointment | 39 | 16.5 |
| Referred | 109 | 46.2 |
| Self-refer | 88 | 37.3 |

post-visit variables, the type of the disease, the number of patients in the queue, and the type of diagnostic tests were factors associated with the overall time (S4C Table).

**Multivariable linear regression analysis result.** For patients who were referred the mean overall time was 52.03 (95%CI 21.65, 82.412) minutes higher than for patients who were an appointment. When the patient number at the queue increased by one the overall time also increased on average by 4.65(95%CI 3.983, 5.324) minutes. For patients who had the diagnostic tests (other Dxic tests), the mean overall time was 96.43(95%CI 52.076, 140.787) minutes higher than patients who had not had any test performed.

Overall time = 120.28minutes + 4.65 (# of patients at the queue) minutes + 52.03 (type of visit (referred) minutes + 96.43 (diagnostic test (other diagnostic tests) minutes (Table 6).

Adjusted $R^2$ was 0.522 but the value of $R^2$ was 0.534 therefore; the variables in the multivariable model explain 53.4% of the variance of the overall time. That means a 53.4% variation in the overall time was due to the difference in the purpose of the visit (referred), the presence of diagnostic tests, and the number of patients in the queue.

## Discussion

The average overall time in this study was 312.3 minutes (5.2 hours). The laboratory unit had the longest average overall time. The pharmacy unit had the shortest average overall time. The average wait time was 213.9 minutes (3.57 hours), and the average consultation (service) time-was 50.3 minutes (0.838 hours). The number of patients in the queue, the type of visit (referred), and the type of diagnostic test (other diagnostic tests) were the factors that determined overall time.

**Table 3. Post-visit characteristics of the respondent in the outpatient unit of Jimma zone hospitals in southwest Ethiopia, 2018 (n = 236).**

| Post-visit characteristics of the respondent | Frequency | Percent |
|---|---|---|
| **Arrival time** | | |
| Before 8am | 5 | 2.12 |
| 8-9am | 119 | 50.4 |
| 9-10am | 63 | 26.7 |
| 10-11am | 11 | 4.7 |
| 11-12am | 2 | 0.85 |
| After 12am | 36 | 15.3 |
| **Date of visit** | | |
| Monday | 40 | 16.9 |
| Tuesday | 41 | 17.4 |
| Wednesday | 40 | 16.9 |
| Thursday | 73 | 30.9 |
| Friday | 42 | 17.8 |
| **Duration in hospital (perceived response)** | | |
| Long | 125 | 52.9 |
| Fair | 77 | 32.6 |
| Short | 34 | 14.4 |
| **Point of delay (perceived response) \*\*** | | |
| Registration | 15 | 6.4 |
| Triage | 13 | 5.5 |
| Examination | 82 | 34.7 |
| Laboratory | 113 | 47.9 |
| Other diagnostics | 14 | 5.9 |
| Pharmacy | 5 | |
| **Reason for delay (perceived response) \*\*** | | |
| Many patients | 135 | 57.2 |
| Staff failed to respond timely | 90 | 38.1 |
| Jumping of the queue by staff | 5 | 1.93 |
| Weak communication | 8 | 3.4 |
| Distance between registration andOPD | 5 | 2.12 |
| Some medicine and laboratory not found here | 16 | 6.8 |
| **Presence of diagnosis** | | |
| Yes | 233 | 98.7 |
| No | 3 | |
| **Type of diagnosis confirmed** | | |
| AFI | 20 | 8.5 |
| UTI | 30 | 12.7 |
| Respiratory disorder | 51 | 21.6 |
| GIT disorder | 40 | 16.9 |
| CVD | 18 | 7.6 |
| Surgical/orthopedics | 37 | 15.7 |
| Skin disease | 11 | 4.7 |
| Cancer | 5 | 2.15 |
| Others\*\*\* | 21 | 9.01 |

\*\*For delay unit and reason of delay the response was multiple choice so the total percentage was greater than 100.

\*\*\* presents dental cases, ophthalmic cases, gynecology and mental cases.

**Table 4. Waiting time (in a minute) at each section of OPD in Jimma zone hospitals in southwest Ethiopia, 2018 (n = 236).**

| | | Mean (SD) | Median | Maximum | Minimum |
|---|---|---|---|---|---|
| Registration | | 19.22(±25.6) | 10 | 246 | ** |
| Triage | | 22.2(±20.7) | 18 | 130 | 2 |
| Examination | | 63.9(±43.73) | 58 | 300 | 1 |
| Laboratory | Pre | 68.6(±39.9) | 56 | 205 | 10 |
| | Post | 102.4(±44.7) | 95 | 290 | 27 |
| x-ray | Pre | 78.8(±46.12) | 68 | 163 | 15 |
| | Post | 11.3(±19.8) | 3 | 81 | 1 |
| Other Dxics*** | | 64 (±65.6) | 49.5 | 333 | 2 |
| Pharmacy | | 6.22(±6.9) | 5 | 57 | ** |

**presents zero waiting time (after arriving immediately gained the services).

*** presents ultrasound, sputum examination& FNA/C.

**Table 5. Service time (minutes) within the different sections of OPD of Jimma zone hospitals in southwest Ethiopia, 2018 (n = 236).**

| Units | Mean (SD) | Median | Maximum | Minimum |
|---|---|---|---|---|
| Registration | 9.4(±6.4) | 8 | 55 | 2 |
| Examination | 25.2(±14.9) | 22 | 105 | 4 |
| Laboratory | 5.35(±2.8) | 5 | 17 | 2 |
| x-ray | 9.2(±1.23) | 9 | 12 | 7 |
| Other Dxics*** | 50(±52.4) | 30 | 210 | 10 |
| Pharmacy | 8.6(±5.4) | 7 | 32 | 2 |

*** presents ultrasound, sputum examination & FNA/C.

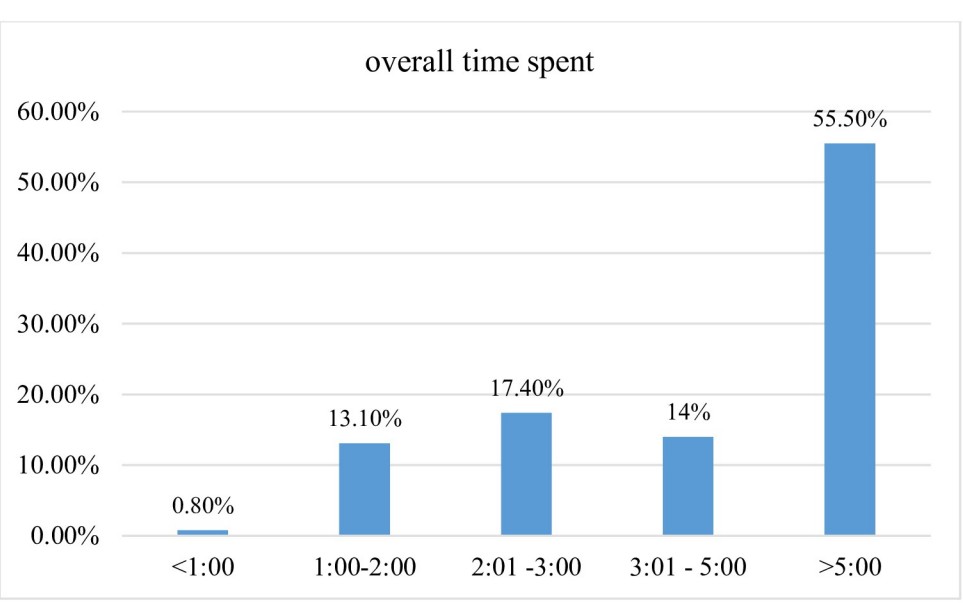

**Fig 1. The overall time in hours patients spent from entry to exit in OPD of Jimma zone hospitals, southwest Ethiopia in 2018 (n = 236).**

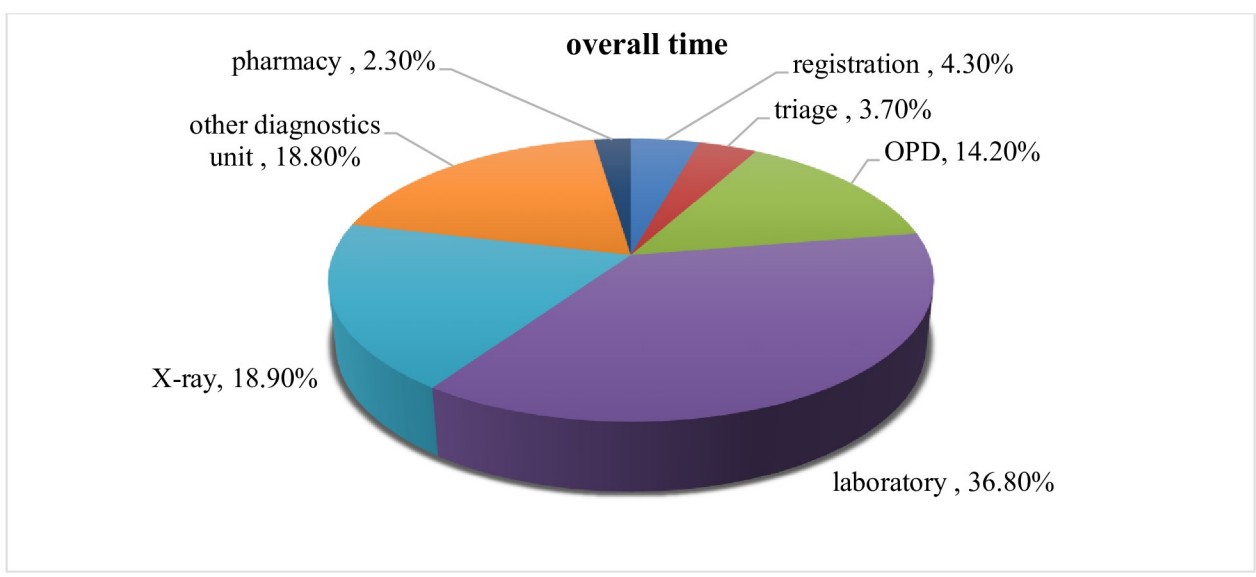

**Fig 2. The overall time in per cent patients spent at each section of OPD, Jimma zone hospitals in southwest Ethiopia, 2018.** The mean (SD) overall time, the patient spent in JUMC, Agaro general hospital, and Seka primary hospital was 336.64 ± 158.72minutes, 275.58 ±144.1minutes and 173.86 ±115.76 minutes respectively (S3 Table).

This study found that the overall median and mean time were 342.5 minutes (approximately 5.7 hours) and 312.33 minutes respectively. This is in line with a study from India that found a median length of stay was 302 minutes [23]. However, this result was higher than the study conducted in the GOPD of a teaching hospital in northwestern Nigeria and in the OPD of NigestEleni Mohammed Hospital, Hossana, where the mean length of stay was 168 minutes and 122.2 minutes, respectively [8, 24]. This could be because the appointment system for

**Table 6. The final model fit variables for overall time spent by patients of OPD in Jimma zone hospitals in southwest Ethiopia, 2018 (n = 236).**

| model | Unstandardized Coefficient (B) | 95%confidence interval for B | Collinearity statistics | |
| --- | --- | --- | --- | --- |
| | | | VIF | Tolerance |
| Constant | 120. 281 | 89.9 150.662*** | | |
| Number of patients in the queue | 4.653 | 3.983 5.324*** | 1.054 | 0.949 |
| Purpose of visit(referred) Review (**reference**) | 52.031 | 21.65 82.412** | 1.137 | 0.88 |
| Diagnostic test (other diagnostics****) No test (**reference**) | 96.431 | 52.076 140.787*** | 1.112 | 0.899 |
| Date of visit(Wednesday) Monday (**reference**) | -17.586 | -55.797 20.626 | 1.019 | 0.982 |
| Disease type (GIT) | 2.09 | -36.592 40.773 | 1.044 | 0.958 |
| Disease type(surgical) Skin problem (**reference**) | -32.506 | -74.584 9.571 | 1.16 | 0.862 |

** = P< 0.005,

*** = P <0.001 (Means statistical significant at stated P-value); R = 0.731, R2 = 0.534, adjusted R2 = 0.522

**** presents ultrasound, sputum examination& FNA/C

follow-up patients is not staggered which means that many patients arrive at the same time on the same day, healthcare providers are late and pay little attention to punctuality, high patient load, jumping of a queue by patients or staff members, waiting of laboratory (other investigation) result and physicians to see the laboratory results and some services are not provided [8, 24]. In addition, in developing countries, the ratio of patients to physicians is very high, which means unable to meet the recommendation of IOM which stated as at least 90% of the patients should be seen within 30 minutes of their appointment time. In Ethiopia, the ratio is currently 15,000:1, but the WHO recommendation was 1000:1 [8, 20].

This study found that the mean overall time for the pediatric and adult patients was 231.02 (±153.9) minutes and 383.31(±129.2) minutes respectively. This finding was higher compared to the study done in Malawi which was 134.9(±65.5) minutes and 110.7(±67.9) minutes respectively. This difference might be because JUMC is a teaching and referral hospital which means most patients were referred patients and professionals were not arrive timely and exposed patients spent a long time at the assessment centre. But the mean overall time between the pediatric and adult age groups in this study was different from the above study. This variation might be, as observed during the study most pediatric patients ended up at examination with a prescription of medicine [22].

This study found that the mean total waiting time was 214 minutes (3.57 hours). This finding was higher compared to the study done in Malaysian and US tertiary hospitals with a mean time of >2hrs and 1:30–3:00 respectively [9, 15]. As known the difference between the study area setting and one of the hospitals (JUMC) is teaching and referral, which leads professionals to fail to respond timely because related to the morning session, professionals do not arrive timely to the OPD which increases waiting time and patient load. Additionally, being referred patient and the absence of some services in the hospital causes the patient to go out to get service and came back to the hospital during this time the professional may not also be present. As observed in the study some patients spent 3–4 hours in the afternoon without any service because some physicians were absent in the afternoon.

The mean waiting time at examination (OPD) was 63.9 +43.7minutes. This finding is higher compared to the study done at OPD in Hosanna which was 30.9+18.4 minutes [24]. This variation might be explained by the fact that this study includes three hospitals and one hospital (JUMC) is a teaching and specialized (referral) hospital due to these professionals not arriving timely and patient load increases.

The mean waiting time at the laboratory was 68.64 minutes + 39.9 minutes (before giving the sample) and 102.36 minutes + 44.7 minutes (after giving the sample) which was approximately similar to the study done on patient satisfaction at JUMC. Similarly, findings showed that the mean time for patients to be x-rayed was 1.655hrs+0.799hrs. This showed a slight difference with the study on patient satisfaction at JUMC which was 1.91+ 0.79hrs. This might be due to the new building and introduction of the citizen charter in 2016 and this study conducted in the new building create conditions easily [25].

Results from this study showed that the mean service time was 50.3 minutes; this finding was higher compared to the study done in Malaysia, China and Indias which were 15 minutes, 17.8 minutes and 13 minutes respectively. This inconsistency is because this study was conducted both on adult and pediatric age groups i.e. in pediatric age groups difficult-to-perform physical examination easily, expose the patient to spending a long time with the provider and one of the study hospitals (JUMC) is a teaching hospital that encourages students to attend patients [9, 15, 26].

The mean adult examination (OPD) time was 22.3 minutes this finding was lower than the time stated at Jimma zone public hospital citizen charter which was 45 minutes [19]. This difference may be due to the high patient load and staff (physicians) not arriving timely resulting

in an increased patient flow rate that means to give service to all waiting patients the physicians may encourage patients to finish their examinations within a short time which decreased adult OPD service time. But this result was higher compared to the study done in OPD of Nigeria tertiary hospital which was 7 minutes [8]. This variation might be due to as known the study including teaching and referral hospital and observed during the study most of the professionals in the OPD room were interns who allow patients to spend a long time with them; most follow-up patients are seen by different physicians, and presence of week communication.

While the mean examination time for pediatric age was 28.5 minutes this finding was higher compared to the time stated at Jimma zone public hospitals citizen charter which was 20 minutes [19]. This can be due to pediatric age groups being unable fully express their feelings, so to capture the problem that not explain by patients the physicians perform deep physical examinations. In addition, difficult to perform physical examinations easily in pediatric age groups, especially less than five years.

Additionally median service time at registration, laboratory, x-ray and pharmacy were 8 minutes, 5 minutes, 9 minutes and 7 minutes respectively. This finding was approximately similar to Jimma zone public hospitals charter stated time which was 5 minutes, 10 minutes, 10 minutes and 5 minutes respectively [19].

Results showed that the number of patients in the queue significantly affect the overall time patient spent at the assessment centre which means the mean overall time increased by 4.65 (95%CI 3.983, 5.324) minutes as the patient number increased by one unit. This result was supported by the study done in Malaysia at Selangor and North West Nigeria [8, 20]. The possible explanation might be due to the large fee differential between public and private health care services. In addition to this; most walk-in cases are coming to public hospitals expecting better services. Health professionals were not punctual, patients with minor cases came with a walk-in and as known one of the hospitals is a referral and provides service for three regional states, this all caused increases in the patient load.

The other factor was the type of visit (referred) that the patient came to the assessment centres, as the result showed the mean overall time for patients who were referred from other facilities was 52.03 (95%CI 21.65, 82.412)minutes higher than for patients who were an appointment. The possible explanation might be due to patients who were referred needing further investigation than appointment patients. That means depending on the type of the case needs different investigation that prolongs overall time, in addition to this; the absence of some investigations and medicines such as ECHO, Helicobacter pylori test, and medicine for acute ton silo pharyngitis that exposes the patient to going outside and return to the hospital, those all increase the overall time patient spent on the assessment centre.

The third factor was the type of diagnostic test (other diagnostic tests), for patients who went to other diagnostic test units (ultrasound, sputum examination, and FNA/C) the mean overall time was 96.43(95%CI 52.076, 140.787) minutes higher than patients who had no any test performed. The possible explanation might be due to most patients were referred that need further investigation/diagnostic test depending on the type of cases, which coupled with a high patient load increase the overall time patient spent.

## Limitation of the study

The study has certain limitations. First, the arrival times of patients who arrived earlier than opening hours were self-reported. This problem was minimized by having one data collector come earlier to the opening hours. Second, since this study was observational, they needed to follow each movement of the patient from entry to exit and the patient could have changed their Behaviour which brings the hawthorn effect. To solve this problem, trained data

collectors maintained a fairly wide distance to avoid noticing. Third, loss of follow-up which means if service extends to the next day the patient was left causes cancellation of this patient which means a waste of time. This problem was minimized by sociable approaching the patient and bringing their cell phone number. Finally, other factors that affect overall time were not assessed due to resource and time limitations.

## Conclusion and recommendations

In conclusion, we have found that the majority of the patients experience long overall times during their visit to the outpatient department of Jimma zone public hospitals with the greatest time spent waiting to receive services. This overall time was highly associated with the huge number of patients; the majority of them are direct walk-ins and referred respectively. Most delays are identified at the examination, laboratory and other diagnostics units. These delays could be attributed to the long queues at OPD, laboratory and other diagnostics service points and staff failure to respond timely. Most of the patients have conditions that can be handled at lower health facilities thus increasing the burden for the hospital to provide quality care for those who have been referred to the referral hospital.

As recommendations, first, the hospital administrations should have strong follow-ups on health professionals both in the morning and afternoon and create a solution in the morning session. Second, the laboratory was the most crowded area so the hospital administration should solve this problem. Since most patients are concentrated on during the start (Monday) and end (Friday) of the week, hospital administration should apply a more effective scheduling system that means scheduling appointments according to expected consultation time especially in the management of chronic conditions or in circumstances where there is a link between the health centre and other hospitals, especially with facilities located at Jimma town.

To reduce the number of patients in the queues, the hospital should work hand in hand with the ministry of health should enforce a policy that was well understood and embraced by all health workers in the lower health facilities. And the hospital administration works to reduce the number of patients in the queues by strengthening the triaging system to enable the screening of patients with minor illnesses. Furthermore, most patients with minor illness queue with those with more complicated illnesses. Finally, recommend for researchers investigate further on quality of service.

## Supporting information

**S1 Table.** A. Waiting time (in a minute) at each section of OPD in Jimma zone public hospitals 2018.(n = 236). B. The total waiting time the patient spends in OPD of Jimma zone public hospitals 2018. (236).
(ZIP)

**S2 Table.** A. Service times (minutes) within the different sections of OPD in Jimma zone public hospitals 2018. (n = 236). B. The service time in minutes based on the type of OPD at Jimma zone public hospitals 2018.(n = 236). C. The total service time the patient spends in OPD of Jimma zone public hospitals 2018. (n = 236).
(ZIP)

**S3 Table. Overall time patient spent from entrey to exit in OPD of Jimma zone public hospitals 2018 (n = 236).**
(DOCX)

**S4 Table.** A. Bivariate linear regression, assessing the association between overall times of patient spent from entry to exit and socio-demographic factors at Jimma zone public hospitals

Southwest Ethiopia, 2018. B. Bivariate linear regression, assessing the association between overall times patient spent from entry to exit and pre-visit factors at Jimma zone public hospitals Southwest Ethiopia, 2018 (n = 236). C. Bivariate linear regression assessing the associations between overall time patients spent from entry to exit in the assessment center and post-visit factors in Jimma zone public hospitals Southwest Ethiopia, 2018.(n = 236).
(ZIP)

## Acknowledgments

We would like to thank the institute of health, Jimma University for allowing the opportunity to conduct the study. We also thank the staffs of Jimma University Medical Center, Agaro general hospital, and Seka primary hospital for providing preliminary data for the development of the proposal in this study. Finally, our gratitude goes to the participants of the study.

## Author Contributions

**Conceptualization:** Zebader Walle, Yibeltal Sraneh, Tilahun Fufa, Elias Ali Yesuf, Gete Berihun.

**Data curation:** Zebader Walle, Dejenie Melese, Tilahun Fufa, Elias Ali Yesuf, Gete Berihun.

**Formal analysis:** Zebader Walle, Yibeltal Sraneh, Elias Ali Yesuf.

**Investigation:** Zebader Walle, Frehiwot Worku, Yibeltal Sraneh, Dejenie Melese, Tilahun Fufa, Elias Ali Yesuf, Gete Berihun.

**Methodology:** Zebader Walle, Frehiwot Worku, Yibeltal Sraneh, Dejenie Melese, Tilahun Fufa, Elias Ali Yesuf, Gete Berihun.

**Project administration:** Zebader Walle.

**Resources:** Zebader Walle, Frehiwot Worku, Yibeltal Sraneh, Dejenie Melese, Tilahun Fufa, Gete Berihun.

**Software:** Zebader Walle, Frehiwot Worku, Yibeltal Sraneh, Elias Ali Yesuf, Gete Berihun.

**Supervision:** Zebader Walle, Frehiwot Worku, Yibeltal Sraneh, Dejenie Melese, Elias Ali Yesuf, Gete Berihun.

**Validation:** Zebader Walle, Frehiwot Worku.

**Visualization:** Zebader Walle.

**Writing – original draft:** Zebader Walle.

**Writing – review & editing:** Zebader Walle, Frehiwot Worku, Yibeltal Sraneh, Dejenie Melese, Tilahun Fufa, Elias Ali Yesuf, Gete Berihun.

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
