## [Decision Letter · Decision Letter 0]

5 Sep 2022

PONE-D-22-02050Overall Time spent by clients from entry to exit and associated factors in out-patient departments in public hospitals of Jimma Zone Southwest, EthiopiaPLOS ONE

Dear Dr. Walle,

Thank you for submitting your manuscript to PLOS ONE. After careful consideration, we feel that it has merit but does not fully meet PLOS ONE’s publication criteria as it currently stands. Therefore, we invite you to submit a revised version of the manuscript that addresses the points raised during the review process.

ACADEMIC EDITOR:

- The abstract is loosely written. It is not as informative as expected. A standard abstract must present, without leaving any doubt, the objective of the paper precisely; source of data (which is not present in your abstract) and analytical approach used; key findings and any policy implication and recommendations.

- I suggest the authors to improve the introduction section. Authors should better highlight the objective of their work and to what extent it contributes to close a gap in the existing literature and/or practice. What is the innovative value of the contribution proposed by the authors?

- You should provide more recent references published in last two-three years in the Literature review. Remove references published before 2018.

- Explain in more details in the data used in the case study, the data for the testing, the criterion for the accuracy, and others to claim these points.

- Discussion section. How should we know about the quality of these solutions? Could you compare these results with some existing approaches in literature? The improvement must be discussed.

- The conclusion section seems to rush to the end. The authors will have to demonstrate the impact and insights of the research. The authors need to clearly provide several solid future research directions. Clearly state your unique research contributions in the conclusion section. Add limitations of the model. No bullets should be used in your conclusion section.

We look forward to receiving your revised manuscript.

Kind regards,

Dragan Pamucar

Academic Editor

PLOS ONE

Journal Requirements:

3. In the ethics statement in the Methods, you have specified that verbal consent was obtained. Please provide additional details regarding how this consent was documented and witnessed, and state whether this was approved by the IRB"

4. Please include additional information regarding the survey or questionnaire used in the study and ensure that you have provided sufficient details that others could replicate the analyses. For instance, if you developed a questionnaire as part of this study and it is not under a copyright more restrictive than CC-BY, please include a copy, in both the original language and English, as Supporting Information.""

5. We note that you included minors (age<18) in your study. Please provide additional details regarding minors consent. In the ethics statement in the Methods and online submission information, please ensure that you have specified whether you obtained consent from parents or guardians. If the need for consent was waived by the ethics committee, please include this information.

“We would like to thank institute of health, Jimma University for funding the study. We also thank staffs of Jimma University Medical Center for providing preliminary data for the development of the proposal in this study. Finally, our gratitude goes to the participants of the study.”

‘The authors received no specific funding for this work”

7. Your ethics statement should only appear in the Methods section of your manuscript. If your ethics statement is written in any section besides the Methods, please move it to the Methods section and delete it from any other section. Please ensure that your ethics statement is included in your manuscript, as the ethics statement entered into the online submission form will not be published alongside your manuscript.

8. We note that you have referenced (Bisanju WR et al. [4]) which has currently not yet been accepted for publication. Please remove this from your References and amend this to state in the body of your manuscript: (ie “Bisanju WR et al. [Unpublished]”) as detailed online in our guide for authors http://journals.plos.org/plosone/s/submission-guidelines#loc-reference-style.

9. Please include a separate caption for each figure in your manuscript

Additional Editor Comments (if provided):

- The abstract is loosely written. It is not as informative as expected. A standard abstract must present, without leaving any doubt, the objective of the paper precisely; source of data (which is not present in your abstract) and analytical approach used; key findings and any policy implication and recommendations.

- I suggest the authors to improve the introduction section. Authors should better highlight the objective of their work and to what extent it contributes to close a gap in the existing literature and/or practice. What is the innovative value of the contribution proposed by the authors?

- You should provide more recent references published in last two-three years in the Literature review. Remove references published before 2018.

- Explain in more details in the data used in the case study, the data for the testing, the criterion for the accuracy, and others to claim these points.

- Discussion section. How should we know about the quality of these solutions? Could you compare these results with some existing approaches in literature? The improvement must be discussed.

- The conclusion section seems to rush to the end. The authors will have to demonstrate the impact and insights of the research. The authors need to clearly provide several solid future research directions. Clearly state your unique research contributions in the conclusion section. Add limitations of the model. No bullets should be used in your conclusion section.

Reviewers' comments:

Reviewer's Responses to Questions

**Comments to the Author**

1. Is the manuscript technically sound, and do the data support the conclusions?

Reviewer #1: Partly

2. Has the statistical analysis been performed appropriately and rigorously? 

Reviewer #1: Yes

3. Have the authors made all data underlying the findings in their manuscript fully available?

Reviewer #1: No

4. Is the manuscript presented in an intelligible fashion and written in standard English?

Reviewer #1: No

5. Review Comments to the Author

Reviewer #1: Date 12/07/2022

To Editorial Board,

PLOS ONE

REVIEW COMMENTS

TITLE: Overall Time spent by clients from entry to exit and associated factors in out-patient departments in public hospitals of Jimma Zone Southwest, Ethiopia

Manuscript number: PONE-D-22-02050

Thank you for inviting me to review this interesting article. Please kindly find the comments provided.

Comments

1. In title, I believe the word ¨overall ¨is not important, why because we are measuring the time the patient spent from entry to exit which by default is the overall time spent by the patient.

2. In Abstract, under background section, researchers should start with the definition of Overall time. its importance in relation to quality of care, and then describe the existing gaps that have led them to conduct this study.

3. In conclusion section of the abstract, the authors stated that "The majority of the patients stayed for an extended period of time”. However, the extended time is not clearly operationalized in this document. The authors have found several numbers of minutes, but their implications are not included.

4. .

"This strongly relates to a high patient load, an absence of some services, being referred patients, and a lack of professional responsiveness". This has not been studied. The authors should interpret the findings of their study and put necessary recommendation based on what they have found. An absence of some services, and a lack of professional responsiveness were not shown to be associated in result section. there is over interpretation of the findings.

5. In introduction section, the authors did not include the standard time that one patient should stay in health care facility. As a result, we cannot determine the implication of overall time spent by the patients. Whether they are below the standard or above the standard time?

This information is very important to forward the recommendation the included hospitals.

The abbreviation should be written in expanded form when they are used for the first time in the text, thereafter the authors can use the abbreviation only. Eg. Outpatient Department (OPD), Jimma University Medical Center(JUMC).

6. The information in line 87-91 should be modified as below or the authors can consider another better alternative.

The length of time a patient spends in the hospital is one indicator of the service quality. In this regard, Jimma University developed a citizen charter in October 2016 to deliver timely services at Jimma University Medical Center (JUMC). However, the overall time that patients spent in JUMC from the time they enter to the time they leave the hospitals has not been studied. Therefore, this study aimed to evaluate the overall time patients spent from entry to exit and associated factors in out-patient departments of public hospitals of Jimma Zone.

But this works if the study setting is only JUMC .

Methods

7. In study setting section the authors should describe hospitals selected for this study in terms of their capacity, major service the hospitals are rendering, and the annual number patient follows up.

8. The issue of validity and reliability of instruments are not addressed and the source from which these instruments were adopted or adapted is not cited.

Result

9. In line 155 ¨R2 ¨ should be R2.

10. The information in lines 156-158 should be described in dedicated result section and removed from method section.

11. In table 1: it is not clear why the number are different from 236 for some variables eg, educational status 172, monthly income 72, employment status 134.

12. In table 2, pre-visit facility characteristics of respondents prior coming to the outpatient units,

This table is dedicated for pre visit facility characteristics; therefore, the table should be about those who had gone to other health care facilities(N=121) prior to coming to the outpatient units the selected hospitals. However, it is not clear why the authors are interested to include the type of visit and frequency of visit for all participants(n=236) in this table.

13. It is not clear why the authors used different abbreviations for the same variables in different table. (Dics*** and Other Dxics***) in table 4 and 5, respectively.

14. The waiting time, service time and overall time should be described based on the types of the hospitals selected for this study.

15. The authors did not include the results of bivariate analysis.

Discussion,

16. In the result section, the overall time was described in terms of the minutes, however, the discussion section the overall time is presented by hours. It should be consistent throughout sections.

17. In line 233" Indies" should be India

18. Information in lines 236-240 need to be cited.

19. Even though the authors have included three hospitals, when they justify the observed differences, they used JUMC only, why the other included hospitals are ignored in the discussion?

20. In lines 255-259, did the authors assess those variables? professionals’ not arrive timely to the OPD?? Did you assess this?

21. In line 317 it is not clear when to say long overall times as we do not have the standard in this document. This should have been operationalized in method section.

22. In this study authors used 3 hospitals that have significant difference in terms of the level of care, their type, number of patient flow, administration and etc. in this case it is not clear to which group of patients they want to generalize the results of their study.

23. Overall, the discussions are not adequate and the results are interpreted in terms of one institution (JUMC) ignoring the other two included hospitals.

24. In abbreviation section

FNA/C, GOPD, and IOM should be added

References

25. Some references are incomplete. Please thoroughly revise them. eg,. #20

26. Additionally, the manuscript has serious grammatical and language problems.

6. PLOS authors have the option to publish the peer review history of their article (what does this mean?). If published, this will include your full peer review and any attached files.

Reviewer #1: No

---

## [Author Response · Author response to Decision Letter 0]

5 Dec 2022

Editor comment. Please include additional information regarding the survey or questionnaire used in the study and ensure that you have provided sufficient details that others could replicate the analyses. For instance, if you developed a questionnaire as part of this study and it is not under a copyright more restrictive than CC-BY, please include a copy, in both the original language and English, as Supporting Information.""

Response: thank you for your concern. The questionnaire was adapted by reviewing different literatures, for more information all things regarding to the tool incorporates in the method part of the revised manuscript (see it).

Reviewer comment. In table 1: it is not clear why the number are different from 236 for some variables eg, educational status 172, monthly income 72, employment status 134.

Response: thank you very much for your comment. Our study included both pediatric and adult age groups as study participants that means those pediatrics age groups create variation in educational status (children with age of below education unable to classify as “unable to read and write” rather we left them as its) and employment status (according to our country labor organization the minimum age to start work is 14 so individuals with age of below 14 left as its because we cannot said unemployment). For income its below the employment status because for “house wife” and “unemployed” individuals there was no any monthly income.

---

## [Decision Letter · Decision Letter 1]

30 Jan 2023

PONE-D-22-02050R1Overall Time spent by clients from entry to exit and associated factors in out-patient departments in public hospitals of Jimma Zone Southwest, EthiopiaPLOS ONE

Dear Dr. Walle,

Thank you for submitting your manuscript to PLOS ONE. After careful consideration, we feel that it has merit but does not fully meet PLOS ONE’s publication criteria as it currently stands. Therefore, we invite you to submit a revised version of the manuscript that addresses the points raised during the review process. Please submit your revised manuscript by Mar 16 2023 11:59PM. If you will need more time than this to complete your revisions, please reply to this message or contact the journal office at plosone@plos.org. Please include the following items when submitting your revised manuscript:A rebuttal letter that responds to each point raised by the academic editor and reviewer(s). You should upload this letter as a separate file labeled 'Response to Reviewers'.A marked-up copy of your manuscript that highlights changes made to the original version. You should upload this as a separate file labeled 'Revised Manuscript with Track Changes'.An unmarked version of your revised paper without tracked changes. You should upload this as a separate file labeled 'Manuscript'.

We look forward to receiving your revised manuscript.

Kind regards,

Dragan Pamucar

Academic Editor

PLOS ONE

Reviewers' comments:

Reviewer's Responses to Questions

**Comments to the Author**

1. If the authors have adequately addressed your comments raised in a previous round of review and you feel that this manuscript is now acceptable for publication, you may indicate that here to bypass the “Comments to the Author” section, enter your conflict of interest statement in the “Confidential to Editor” section, and submit your "Accept" recommendation.

Reviewer #1: (No Response)

Reviewer #2: (No Response)

2. Is the manuscript technically sound, and do the data support the conclusions?

Reviewer #1: Yes

Reviewer #2: Yes

3. Has the statistical analysis been performed appropriately and rigorously? 

Reviewer #1: Yes

Reviewer #2: Yes

4. Have the authors made all data underlying the findings in their manuscript fully available?

Reviewer #1: No

Reviewer #2: Yes

5. Is the manuscript presented in an intelligible fashion and written in standard English?

Reviewer #1: Yes

Reviewer #2: No

6. Review Comments to the Author

Reviewer #1: I need more clarification on why some comments are not addressed . Therefore, I herewith attached the same comments.

Reviewer #2: Thank you for making a revision and comments to the reviewer. The manuscript seems to be improved. However, it seems to have still some concerns.

Major comments

1. Abstract: The sentences in the Conclusion and Recommendation section are still overstatements. Even though the data in the supplementary tables can support the sentences, the statements cannot be concluded by the contents of the abstract. I think that abstract should be independent and self-sufficient.

2. Results and Discussion: Although the authors addressed the previous Questions from the former reviewer, such as Questions #4 and #21, by adding the information as the supplementary tables, I think it is not enough. If the authors want to insist that "lack of professional responsiveness" or the specific character of the referral hospital (e.g., most physicians are interns) could be the main reasons for the long overall time, the authors should put the data in the main document rather than providing the data in the supplementary files.

Minor comments

3. Table 6. please provide the references for each variable as shown in the supplementary files.

7. PLOS authors have the option to publish the peer review history of their article (what does this mean?). If published, this will include your full peer review and any attached files.

Reviewer #1: No

Reviewer #2: No

---

## [Author Response · Author response to Decision Letter 1]

29 Mar 2023

Question: #1 I need more clarification on why some comments are not addressed. Therefore, I here with attached the same comments.

Response: sorry for missing some comments in the previous revision. We have incorporated it in the revised version of manuscript.

Response to reviewer 2

Q#1: Abstract: The sentences in the Conclusion and Recommendation section are still overstatements. Even though the data in the supplementary tables can support the sentences, the statements cannot be concluded by the contents of the abstract. I think that abstract should be independent and self-sufficient.

Response: thank you for your insight. We have incorporated it based on your request in the revised version of the manuscript. 

Q #2: Results and Discussion: Although the authors addressed the previous Questions from the former reviewer, such as Questions #4 and #21, by adding the information as the supplementary tables, I think it is not enough. If the authors want to insist that "lack of professional responsiveness" or the specific character of the referral hospital (e.g., most physicians are interns) could be the main reasons for the long overall time, the authors should put the data in the main document rather than providing the data in the supplementary files.

Response: thank you for your suggestion. In the main document of table 3 there is the reason of delay (long overall time) in the hospitals. From the reasons “professionals not response timely or late” was the one reason. This may be happen because of the one selected hospital is referral and teaching that has morning session for intern students whom respond in most OPD of the assessment center.

Q#3: Table 6. Please provide the references for each variable as shown in the supplementary files.

Response: thank you for your comment we have corrected it in the revised version of the manuscript.

---

## [Decision Letter · Decision Letter 2]

18 Dec 2023

Overall Time spent by clients from entry to exit and associated factors in out-patient departments in public hospitals of Jimma Zone Southwest, Ethiopia

PONE-D-22-02050R2

Dear Dr. Walle,

We’re pleased to inform you that your manuscript has been judged scientifically suitable for publication and will be formally accepted for publication once it meets all outstanding technical requirements.

Kind regards,

Dragan Pamucar

Academic Editor

PLOS ONE

Additional Editor Comments (optional):

The authors have addressed the point of my concern. I am happy with their corrections. Hence, I would like to recommend this manuscript to be published.

Reviewers' comments:

Reviewer's Responses to Questions

**Comments to the Author**

1. If the authors have adequately addressed your comments raised in a previous round of review and you feel that this manuscript is now acceptable for publication, you may indicate that here to bypass the “Comments to the Author” section, enter your conflict of interest statement in the “Confidential to Editor” section, and submit your "Accept" recommendation.

Reviewer #1: All comments have been addressed

Reviewer #2: All comments have been addressed

2. Is the manuscript technically sound, and do the data support the conclusions?

Reviewer #1: Yes

Reviewer #2: Partly

3. Has the statistical analysis been performed appropriately and rigorously? 

Reviewer #1: Yes

Reviewer #2: Yes

4. Have the authors made all data underlying the findings in their manuscript fully available?

Reviewer #1: Yes

Reviewer #2: (No Response)

5. Is the manuscript presented in an intelligible fashion and written in standard English?

Reviewer #1: Yes

Reviewer #2: (No Response)

6. Review Comments to the Author

Reviewer #1: Thank you for addressing the previous comments. With regard to the model you have utilized, one of the assumptions of linear regression is that the dependent variable should be continuous and normally distributed. In your case, the dependent variable is the time, which is mostly not normally distributed. How did you address this issue, or what is your explanation for using linear regression?

Reviewer #2: (No Response)

7. PLOS authors have the option to publish the peer review history of their article (what does this mean?). If published, this will include your full peer review and any attached files.

Reviewer #1: No

Reviewer #2: No

---

## [Editor Report · Acceptance letter]

17 Feb 2024

PONE-D-22-02050R2 

PLOS ONE

Dear Dr. Walle, 

I'm pleased to inform you that your manuscript has been deemed suitable for publication in PLOS ONE. Congratulations! Your manuscript is now being handed over to our production team.

Kind regards, 

on behalf of

Dr. Dragan Pamucar 

Academic Editor

PLOS ONE